# The Effects of Astaxanthin on Cognitive Function and Neurodegeneration in Humans: A Critical Review

**DOI:** 10.3390/nu16060826

**Published:** 2024-03-14

**Authors:** Callum James Joshua Queen, S. Andy Sparks, David C. Marchant, Lars R. McNaughton

**Affiliations:** Department of Sport and Physical Activity, Edge Hill University, Ormskirk L39 4QP, UK; queenc@edgehill.ac.uk (C.J.J.Q.); andysparks76@gmail.com (S.A.S.); marchand@edgehill.ac.uk (D.C.M.)

**Keywords:** nutrition, supplement, antioxidant, cognitive enhancement, executive function, cognition

## Abstract

Oxidative stress is a key contributing factor in neurodegeneration, cognitive ageing, cognitive decline, and diminished cognitive longevity. Issues stemming from oxidative stress both in relation to cognition and other areas, such as inflammation, skin health, eye health, and general recovery, have been shown to benefit greatly from antioxidant use. Astaxanthin is a potent antioxidant, which has been outlined to be beneficial for cognitive function both in vitro and in vivo. Given the aforementioned promising effects, research into astaxanthin with a focus on cognitive function has recently been extended to human tissue and human populations. The present critical review explores the effects of astaxanthin on cognitive function and neurodegeneration within human populations and samples with the aim of deciphering the merit and credibility of the research findings and subsequently their potential as a basis for therapeutic use. Implications, limitations, and areas for future research development are also discussed. Key findings include the positive impacts of astaxanthin in relation to improving cognitive function, facilitating neuroprotection, and slowing neurodegeneration within given contexts.

## 1. Introduction

Dietary antioxidants are substances which are present in food or supplements that decrease the adverse effects associated with both reactive nitrogen species (RNS) and reactive oxygen species (ROS) to a significant degree in relation to normal physiological functions [1]. The negative effects of RNS and ROS are thought to contribute significantly to oxidative stress, which is a key factor underpinning cognitive decline and impaired cognitive functions [2]. As a result, nutritional interventions which utilise antioxidants have been of interest when considering methods to reduce cognitive function decline and improve cognitive performance [3]. Common dietary antioxidants include ascorbic acid (vitamin C), α-tocopherol (vitamin E), polyphenols, selenium, and carotenoids [4,5]. There are a wide variety of dietary sources of antioxidants, such as plants, fruits, and vegetables. They can either be ingested as part of a balanced diet or be taken as a supplement. The latter has been of particular interest due to the role supplements have in both the improvement and maintenance of overall health and well-being, especially in individuals where sufficient intake through ordinary diet is challenging. This has been evidenced by the rising prevalence of supplement use in the wider population [6]. Many types of antioxidants have received extensive research attention, particularly ascorbic acid [7,8] and α-tocopherol [9,10]. The carotenoids, which are the antioxidants responsible for producing the bright colours associated with plants, algae, vegetables, and fruits, have received some research attention, which, when considered holistically, covers a wide variety of applications [11,12,13]. A key contributing factor towards the depth of this research base is accessibility, as carotenoids are typically obtained from readily available fruits, vegetables, and supplements. They can also protect cells from oxidative processes mediated by either singlet oxygen, free radicals, or light [14]. Due to the absence of accessibility issues surrounding ingestion and the outlined advantageous properties, carotenoids have been investigated as a beneficial area of exploration [15,16].

One of the most potent carotenoids is natural astaxanthin (AST) [17], which has been shown to offer a superior radical absorbance capability when compared to antioxidants of a similar chemical nature, such as α-tocopherol [18]. It has been suggested to have the greatest oxygen radical absorbance capability, bettering other antioxidants, such as vitamin E [17,19]. Astaxanthin is found across a range of sources, especially in crustacean and salmonid aquaculture. It is also often acknowledged for generating the unique pink colour associated with salmon, shrimp, krill, and lobster. This colour is thought to stem from 11 conjugated double bonds and how they are expressed [20]. Notable key natural sources of astaxanthin include red yeast, *Phaffia rhodozyma*, shrimp, salmon, crustacean by-products, and some forms of green algae [21].

The specific properties of AST allow it to maintain the structure and subsequent integrity of cell membranes, facilitating both improvement in gene expression and immune system functioning, respectively [22]. The previously established effects are achieved by managing lipid peroxidation (LPO), neutralising ROS, and scavenging free radicals [23]. The primary benefit of AST is, however, its capability for mitigating oxidative stress (OS), both in vitro and in vivo [24]. The OS reduction effect is thought to stem from AST’s interaction with the phosphoinositide 3-kinase/protein kinase B pathway (Figure 1). This interaction has been outlined to facilitate OS reduction due to its role in aiding the process of dissociating NRf2 from KEAP1 [25].

When distinguishing the efficacy of AST from other carotenoids, it is also important to consider the transportation mechanisms adopted by different subgroups. Polar carotenoids, such as zeaxanthin and lutein, are transported using different densities of lipoproteins (high and low). In contrast, non-polar carotenoids, for example, lycopene and β-carotene, are transported by lipoproteins of low and very low density. Astaxanthin has been shown to use all the lipoprotein densities, with low density (29%), very low density (36–64%), and high density (24%) all being outlined as methods of transportation [26]. This range of transportation routes, along with AST’s ability to cross the blood–brain barrier [27] and act both inside and on the surface of the double layer cell membrane [28], as depicted in Figure 1, are thought to contribute highly to its efficacy. This ability offers a clear advantage when contrasted against vitamin C and β-carotene, which individually only act solely either outside or inside of the double layer cell membrane [28].

## 2. Proposed Benefits of Astaxanthin

The benefits of astaxanthin, particularly in human samples and human tissue, have only been explored more thoroughly in recent years [17,29]. This could be suggested to stem from the current focus on dietary supplementation for the general populous as opposed to only athletes [30]. The outlined viewpoint in addition to its economic implications [31] offers a justification to expand on the finding that humans cannot synthesise astaxanthin naturally, meaning it cannot be utilised without being consumed as part of a person’s diet [32]. Astaxanthin supplementation has been used as an intervention in both animal- and human-based studies [17,33,34]. The proposed benefits have been suggested to include, but are not limited to, decreases in inflammation, the combatting of neurodegenerative disease, improvements to cognitive function, anti-cancer properties, better recovery, and improved cellular, eye, skin, and heart health [28,35,36,37].

As more research is being published, the exploration of astaxanthin has progressed away from animal studies and towards human populations. Whilst other proposed benefits of AST, for example, skin health [29], have been reviewed utilising solely human-based research, at present, there has been no critical review outlining the effects of astaxanthin on cognitive function and neurodegeneration holistically in human samples. This offers a rationale for research in the area. The following critical review aims to explore this gap in the wider literature base and therefore aid in furthering overall understanding of the effects of astaxanthin in humans. To explore the effects of AST in relation to cognition and neurodegeneration in its entirety, the effects of astaxanthin as part of a compound, as well as the literature outlining its impact on indirect factors related to cognition, such as mental and physical fatigue, will also be reviewed.

## 3. The Effects of Astaxanthin on Cognitive Function in Humans

Astaxanthin has been researched in relation to various facets of cognitive function in humans, including episodic memory (visual and verbal stimuli), working memory/short-term memory, processing speed, response inhibition, cognitive shifting, and attention [38,39,40]. Research has also been conducted which focusses on the effects of AST in relation to neurological protection and the prevention of neurological degeneration/neurological disease [27,41]. Finally, AST’s effects on cognition as part of a compound, for example, its effects when ingested alongside tocotrienol or sesamin, have also been explored. This area encompasses the effects of AST on indirectly associated factors, such as mental and physical fatigue [42,43]. Whilst AST has been suggested to be beneficial, it is important to review the research critically, as the finalised claims could have implications in the treatment of cognitive impairment/neurodegeneration as well as the enhancement of cognitive function. Furthermore, the outlined implications could contribute significantly to both quality and length of life [44,45]. This is especially the case for cognition due to the established link between oxidative stress and the areas of neurodegeneration, cognitive ageing, cognitive decline, and cognitive longevity [46]. As a result, it is important that research in this area, which has the potential to be used as a basis for change, is both dependable and of a high calibre.

A study examining differences in cognitive function utilised AST supplementation versus a placebo over an 8-week period in a dose of 8 mg·day^−1^ [38]. The research method allowed for different facets of cognition to be assessed. This was achieved using a word memory test, Stroop test (multiple steps), and verbal fluency test. No tests showed any inter-group differences between the placebo and AST groups. However, a further analysis splitting the sample by age (<55 vs. ≥55) revealed significant intra-group findings. The results of the word memory test, which were specifically looking at “words recalled after five minutes”, showed a significant improvement (*p* = 0.027) following AST supplementation in the <55 years subgroup; this indicates that this demographic may experience positive changes to cognitive function following a course of AST supplementation.

Closer scrutiny of the research in this area suggests it is important to consider the role of age, blood pressure (BP), and body mass index (BMI). With age, the prevalence and severity of cognitive performance decline across different areas has been suggested to increase [47]. This decline is thought to stem from a decrease in executive function [48], processing speed [49], or a combination of both [50]. The above relationship in addition to the association between high BP, high BMI, and cognitive decline risk [51] could impact findings in older and unhealthy middle-aged populations. It is also beneficial to acknowledge the potential of the research, as, whilst there were no inter group differences, the AST group showed improvement from their respective baseline measurements on more individual items across the tests when compared to the placebo group. This implies that under different study parameters, which also incorporate a stronger control of extraneous variables, such as age, BMI, and BP, a significant finding could be elicited. More research is therefore needed in this area to substantiate any claims supporting or refuting the efficacy of the supplement. Until such work is conducted, it can be argued that whilst AST does not support these facets of cognition across all age and health demographics, it can offer beneficial effects for cognitive function.

Alternative findings partially support the notion that these results are method- and population-dependent. Elderly subjects (*n* = 96) were given 12 weeks of AST supplementation over the course of a double-blind placebo control trial [39]. Cognition was assessed using both CogHealth, which is a selection of card games completed on a computer that are designed to assess different facets of cognition, and the Groton Maze Learning Test (GMLT—a computer-generated maze requiring individuals to move from one area on the screen to another). Results were compared across two supplementation conditions and a placebo control. The GMLT scores improved across both high- and low-dose conditions (6 and 12 mg·day^−1^), whereas CogHealth scores only improved in the 12 mg·day^−1^ group. It is, however, important to note that these findings, whilst evidenced, were not shown to be statistically significant; therefore, similar research with larger sample sizes, or more robust research designs, would be required to substantiate the results. Further insight can also be gained from a breakdown of the testing methods. Only one of the measures for episodic memory assessed by the CogHealth test improved. The results showed an improvement in working memory; however, no acknowledgeable changes in response time (the second outcome measure associated with visual stimuli), implying that AST only has a partial effect on outcomes of visual episodic memory under given dose-dependent parameters.

There is evidence within the literature to substantiate the claim that AST supplementation can improve working memory. Research by Satoh [52] reported a mean percent accuracy increase from 90.46 to 96.30% (*p* < 0.05 vs. baseline) in a measure of working memory. The findings of the outlined research also provide support for an alternative viewpoint regarding AST and response time. Decreased response time from baseline (*p* < 0.05 vs. baseline) on simple reaction (341.68 to 281.76 ms), choice reaction (504.53 to 463.63 ms), divided attention (494.13 to 412.07 ms), delayed recall (1008.19 to 916.77 ms), and working memory tasks (762.94 to 654.83 ms) was shown following supplementation, suggesting the effects of AST on response time to be task- and population-dependent. Measures of event-related potential (P300) taken during the same study indicated a significant increase (*p* < 0.1 vs. baseline) in amplitude (mean µV: 7.6 to 10.54) despite not showing an increase in latency (mean: 359.40 to 363.10 ms) following 12 weeks of AST supplementation, implying an effect on cognition, memory, attention, and information processing due to the established use of response time as a measure of psychomotor speed [53]. When considering the findings of this research, it is useful to acknowledge the demographic, as the sample did have age-related forgetfulness. Further research substantiating the claim in a completely healthy, or younger, demographic would therefore be advantageous for population-wide generalisability.

The findings of previously outlined research pertaining to the effects of AST suggest that supplementation has no significant effects on measures of response inhibition or cognitive shifting capabilities assessed through semantic and phonemic fluency. The Stroop test has also been used under different parameters to assess changes in response inhibition following AST supplementation [38]. During this study, two outcomes were measured by analysing step two (a reverse Stroop interference task) and step four (the Stroop interference task) of the Stroop test. No significant differences were found in this study (*n* = 54, age = 45–64) within or between groups following 8 weeks astaxanthin supplementation (8 mg·day^−1^).

## 4. The Impact of Astaxanthin on Neurological Protection and the Prevention of Neurological Degeneration/Neurological Disease

Astaxanthin is also thought to be beneficial for neuroprotection [54], as well as aid in the preventing of neurodegeneration [55]. Research both in vivo and in vitro has shown AST to have positive effects on neuronal apoptosis [56]. Neuronal apoptosis can be easily understood as “cell death” and is required for processes such as cell turn over and immune system functioning [57]. However, under negative and uncontrolled pathological conditions, apoptosis is thought to contribute to a selection of human-based cognitive diseases [58]. Astaxanthin is linked to neuronal apoptosis by the underlying mechanisms of oxidative stress. Oxidative stress can lead to the production of ROS and subsequently excitotoxicity stemming from downstream consequences [59]. As a result, neuronal apoptosis may offer a foundation for supplement-based interventions using AST. Further research also offers support for the above pursuit. Loss of the endogenous antioxidant enzymes catalase and superoxide dismutase (SOD) has been shown to be detrimental to cognitive function. It has been suggested that AST counteracts the associated negative effects [27]. This finding is of relevance, as the above molecules/enzymes have also been shown to decrease in efficacy as a person ages; subsequently, their enhancement could play a significant role in both age-associated neurodegenerative disease as well as brain ageing holistically [60,61]. It can therefore also be argued that prevalent neurodegenerative diseases offer another useful avenue for exploring the efficacy of AST in relation to neuroprotection. Conclusions can be drawn from a selection of findings associated with neurodegenerative diseases, such as Parkinson’s disease and Alzheimer’s disease [27,62].

## 5. Astaxanthin and Parkinson’s Disease (PD)

Parkinson’s disease (PD) is a prevalent neurodegenerative condition of unknown cause, which affects susceptible areas of the brain. It is thought to be the result of an interaction between environmental and genetic factors, culminating in progressive neuron degeneration [63]. Research examining the protective effects of AST on apoptosis in the SH-SY5Y human neuroblastoma cell line [64], a cell line associated with PD (following appropriate treatment), infers the carotenoid to be of benefit to the condition. The research showed a pre-treatment targeted at the above cell line with AST suppressed the specific type of monitored apoptosis (6-OHDA-induced). This effect was thought to be the result of AST’s ability to both protect mitochondria and capitalise on its antioxidant potential [65].

These findings are further supported and expanded upon by research using the same cell line [41]. The beneficial effects of AST to the cell line were shown to include a further suppression to endoplasmic reticulum (ER) stress and protection against neuron damage occurring specifically in PD. The AST treatment was suggested to reverse the negative effects of miR-7 knockdown. This builds on the finding that miR-7 has been shown to have decreased concentrations in patients with PD, especially in disease-specific areas of the brain, for example, the substantia nigra [66,67]. Before the research conducted by Shen [41], an established mechanism underpinning AST’s efficacy for the treatment of PD had only been suggested. By building on this finding and conducting research targeting human populations with PD, AST could be identified as an effective therapeutic agent. Until this research is conducted, the above findings cannot be applied with full confidence due to the complex nature of the nervous system [68], which is made up of different cell types, each with multiple functions. Whilst cell lines offer a useful tool for standardised research, they cannot mimic the impacts of a wider environment, which incorporates integrative processes.

## 6. Astaxanthin and Alzheimer’s Disease (AD)

Other neurodegenerative diseases have also been suggested to benefit from AST. Alzheimer’s disease (AD) is a type of dementia best understood through its outcomes, which primarily consist of a reduction in quality/quantity of given neurons, an increase in inflammation-based mediators, and a reduction in acetylcholine [69]. Whilst there are many animal-based studies demonstrating a potentially positive effect of AST on Alzheimer’s disease [70,71], there is minimal evidence to substantiate the claim within humans. Associations can, however, be drawn utilising research focusing on phospholipid hydroperoxides (PLOOH). Abnormally elevated concentrations of PLOOH are thought to manifest within those suffering from specific forms of dementia [72].

A randomised, double-blind, placebo-controlled study [73] looked at the effects of 12 weeks of AST supplementation on PLOOH concentrations in a sample of older adults (*n* = 30, age 50–69 yrs). The sample was divided into three groups: a placebo group, a 6 mg·day^−1^ group, and a 12 mg·day^−1^ group. Both AST supplementation conditions elicited lower PLOOH concentrations than the placebo group at the end of the study period. The placebo group (14·9 pmol·mL packed cells) was shown to differ significantly (*p =* 0.031) from both the 6 mg·day^−1^ (8.0 pmol·mL packed cells) and 12 mg·day^−1^ (9.7 pmol·mL packed cells) AST groups in erythrocyte PLOOH concentrations by the end of the study period. This effect also extended to PLOOH concentrations in the plasma. The results of this study imply that AST may have preventative capabilities for certain forms of dementia such as Alzheimer’s disease, stemming from its effects on the underlying mechanisms. To substantiate this claim, research should be conducted using individuals with a diagnosis of AD. This research should also incorporate a greater selection of doses, as those with AD are acknowledged as having higher concentrations of PLOOH [74], which could impact the amount of AST required to elicit an effect. A further investigation pertaining to study length should also be made. By looking at the potential benefits of AST longitudinally, a greater insight surrounding its effects in those suffering with cognitive impairment could be gained.

## 7. The Effects of Astaxanthin as Part of a Compound on Cognitive Performance

Astaxanthin has also been shown to elicit benefits to cognitive performance when ingested as part of a supplement blend containing additional key ingredients. This offers a potential avenue for exploring the effects of AST, providing the paired ingredients are distinguishable in nature. Research has shown when AST (3 mg·day^−1^) and sesamin (5 mg·day^−1^), which is a lignin capable of ROS scavenging activity [75], are taken simultaneously for a 12-week period, an improvement to psychomotor speed (104.0 to 109.1 CNVS score) and a statistically significant improvement to processing speed (110.0 to 114.9 CNVS score, *p* = 0.018) ensue in those suffering with mild cognitive impairment [76]. The applicability of this research to individuals with mild neurodegenerative symptoms is further highlighted by a double-blind placebo control study, which also showed AST to have a positive effect when taken with members of the vitamin E family (tocotrienols) over a 12-week period [77]. The research was conducted using participants who were deemed healthy but expressed sensations of memory decline (*n* = 36, mean age = 55). Findings from the above study have suggested that supplementation with AST in addition to tocotrienols is of significant benefit to composite memory (*p* = 0.072). When culminated, these findings are of importance, as vulnerable older populations are more susceptible to mild cognitive impairment [78], in part due to the negative association between age and processing speed [79]. As a result, it could be argued that outcomes to various forms of cognitive assessment may differ based on age demographic and the nature of the task being more speed or complexity orientated. It is, however, important to consider that in both aforementioned instances, follow ups separating the effects of AST from the effects of other ingredients within the supplement blend were not performed. Therefore, whilst the underlying mechanisms of the ingredients differ, associations between the effects of AST and cognition proposed by the outlined studies can only be implied.

Astaxanthin has also been investigated in relation to cognitive performance indirectly through fatigue. Elevated levels of both mental and physical fatigue have been negatively associated with various facets of cognitive function [80]. A double-blind, placebo-controlled, parallel group study [42] researched the effects of supplementation with AST (12 mg·day^−1^) and tocotrienol (20 mg·day^−1^) against a control of just tocotrienol (20 mg·day^−1^). This study was conducted on a sample of 39 volunteers all suffering with fatigue induced by mental and physical challenge. The results of the 12-week study found that the AST supplementation condition lowered feelings of fatigue post mental and physical loading to a significantly greater degree than the control condition. This finding was supported by research focussing on the effects of AST and sesamin [43], which showed significantly improved recovery from mental fatigue induced by a four-hour visual display terminal-based task post supplementation (*p* < 0.05). The study also outlined a mechanism for AST’s efficacy, which was a reduction in phospholipid hydroperoxides accumulation. This finding aligns with previously discussed studies [73], therefore substantiating the study claims. Furthermore, the research conducted by Imai [43] also used the same supplement quantities (3  mg·day^−1^ of AST and 5 mg·day^−1^  of sesamin) as previously mentioned studies [76]. This offers support for the protocol and could be indicative of both benefits to processing speed following AST supplementation within shorter time frames, as well as positive synergistic effects.

The previously established effect of AST on fatigue has also been shown to extend to less able and vulnerable demographics. Research surrounding the effects of AST within an obese population found the supplement had positive effects on feelings of fatigue and provided marker specific justification [81]. Compared with a baseline measurement of both malondialdehyde (MDA) and isoprostane (ISP), which are acknowledged markers of lipid peroxidation damage and oxidative stress [82], indices of fatigue and oxidative stress were significantly lower (*p*  <  0.01) across both high- and low-dose groups following three weeks of astaxanthin supplementation. The observed values for MDA changed as follows: 5 mg·day^−1^ mean baseline = 2.71 µmol/L; 20 mg·day^−1^ mean baseline = 2.66 µmol/L to 5 mg·day^−1^ mean three-weeks post = 1.77 µmol/L; 20 mg·day^−1^ mean three-weeks post = 1.72 µmol/L. The observed values relating to ISP also changed as follows: 5 mg·day^−1^ mean baseline = 5.34 ng/mL; 20 mg·day^−1^ mean baseline = 4.63 ng/mL to 5 mg·day^−1^ mean three-weeks post = 1.88 ng/mL; 20 mg·day^−1^ mean three-weeks post = 1.64 ng/mL. Furthermore, the markers for SOD, an important antioxidant enzyme [83], and the total antioxidant capacity (TAC) [84] increased significantly in both supplementation conditions after the three-week period (*p*  <  0.001). This research generates support for the effect of AST on fatigue and provides evidence for an underlying mechanism. It also outlines a potential demographic for therapeutic use.

## 8. Conclusions and Summary of Evidence

Astaxanthin has been shown to benefit different facets of cognitive function in humans, primarily subdivisions of episodic memory, response time, and working memory. Other areas of cognitive function, such as measures of response inhibition or cognitive shifting capabilities, were not shown to be impacted holistically. Across all findings, it is important to consider nuance. Studies which support AST as a supplement for cognitive improvement often use only a particular age demographic, broad cognitive tests, or a small sample size without an appropriate research design. As a result, there is a need for more studies using different populations for similar methods, specific cognitive tests, and more robust research methods, which consistently incorporate control groups or control conditions. Studies not showing a significant benefit from AST supplementation when subdivided further by age have generated discrepancies. More research is therefore also required to decipher if the overarching theme of age as an impacting factor for AST efficacy is routed in correlation, causation, or task specifics, for example, testing methods looking at processing speed.

The research designs of the reviewed studies are important to note in the development of this conclusion. The prevalent testing method CogHealth was originally designed to detect changes surrounding cognition in subjects with both no and mild cognitive impairment [85]. Findings could therefore be less valid due to the test being both non-specific in nature and comparatively easier for non-suffering individuals. To overcome this issue and explore cognitive function thoroughly, initially, a less integrated approach could be taken. Future research could use tests targeting a singular branch of cognition in detail, for example, the OSPAN task, which looks solely at working memory [86]. This specifically targeted format would facilitate a deeper understanding once culminated across both different cohorts and contexts.

The above review also offers a foundation for claims surrounding the effects of AST on neuroprotection and neurodegeneration. Astaxanthin has been highlighted as having a potential therapeutic use for both PD and AD. It is, however, important to consider that within the sphere of neurodegeneration, a strong majority of human-based research is at present still cell-based [41,64]. To substantiate the drawn conclusions about AST and neuroprotection/neurodegeneration, a greater volume of research in healthy human samples or disease-specific human samples would be beneficial. This research should also incorporate key definitions pertaining to the stage and severity of the respective neurodegenerative disease, as it has been acknowledged that there is a long latency period between initial cell damage and the onset of clinical symptoms [87]. It could also be argued that this research should be conducted under similar parameters to establish generalisability [88]. To ensure quality findings moving forward, consistency in the use of robust research method implementation, for example, the use of double-blind placebo control studies, which incorporate a cross over design, should be considered. This suggestion is made based on the strength and reliability of the outlined design [89]. Furthermore, this recommendation could also assist with sample size issues. This specific design could be used to reduce the need for larger sample sizes, as each participant acts as their own control [90].

A further consideration for future research is the standardisation of AST quality. Previous research suggests both AST source and processing method impact supplement standard and therefore potential for eliciting significant effect [27,91,92]. This acknowledgement is of relevance for future studies in which AST is being researched as part of a compound due to the enhanced variability of the overall supplement profile. Finally, the length of trial and dose of astaxanthin should also be standardised. Whilst, at present, the typical dose range within the research sits between 4 and 12 mg·day^−1^, the length of trial/purity of AST dose, and therefore its potency, are not standardised. This makes drawn conclusions between studies of a similar supplemental nature [43,76] less substantial.

The conclusion of the above review is that AST has the potential to improve cognitive function, facilitate neuroprotection, and slow neurodegeneration. This claim is made based on the established positive effects of AST on different branches of memory and response time, as well as the implications of work utilising biomarkers in human populations. There is, however, still a need for further research substantiating the effects of AST on cognitive function and neurodegeneration in humans. Issues such as research method, sample population, cognitive test efficacy, dominance of cell-based findings, dose of AST and study length, supplement quality, and supplement blend distinguishability should be resolved or accounted for before the claims relating to AST and cognition are granted further merit or potentially used as the basis of therapeutic intervention.

## Figures and Tables

**Figure 1 nutrients-16-00826-f001:**
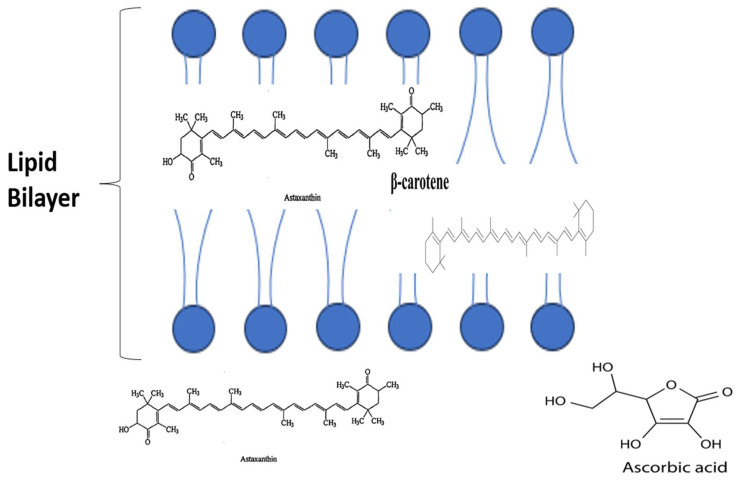
A graphic outlining the breakdown of AST’s actions compared with other carotenoids alongside an illustration of the P13K/Akt pathway and its interactions with subsequent downstream consequences.

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
