# Peer review of "The Effects of Astaxanthin on Cognitive Function and Neurodegeneration in Humans: A Critical Review"

_nutrients, 2024, doi:10.3390/nu16060826_

Round 1

Reviewer 1 Report

Comments and Suggestions for Authors

The authors reviewed the effects of astaxanthin, a potent antioxidant carotenoid, on cognitive function and neurodegeneration in humans. They explain the sources, properties, and mechanisms of astaxanthin, and its potential benefits for various aspects of health and well-being. They discussed the evidence for astaxanthin’s impact on different facets of cognition, such as memory, processing speed, response inhibition, and cognitive shifting. They noted that the results are mixed and depend on factors such as age, dose, and testing method. They suggested that more research is needed to establish the efficacy and generalizability of astaxanthin for cognitive enhancement. The authors examined the role of astaxanthin in neuroprotection and neurodegeneration, focusing on two prevalent diseases: Parkinson’s disease and Alzheimer’s disease. They summarized the findings from cell-based and biomarker studies that indicate astaxanthin’s ability to modulate oxidative stress, inflammation, and apoptosis in neuronal cells. They call for more human-based studies to validate the therapeutic potential of astaxanthin for these conditions. The authors explored the effects of astaxanthin when combined with other ingredients, such as sesamin and tocotrienol, on cognitive performance and fatigue. They reported that astaxanthin may have synergistic effects with these compounds, and may improve recovery from mental and physical stress. They also highlighted the importance of supplement quality and standardization for future research. Specific comments:

1.          How do you justify the need for this critical review? What are the gaps or limitations in the existing literature that you aim to address?

2.          The main body is well-organized and covers a wide range of studies. However, some of the studies have small sample sizes, short intervention periods, or lack of control groups, which may limit the validity and generalizability of the findings. You could discuss these methodological issues and their implications for the interpretation of the results.

3.          How do you balance the positive and negative effects of astaxanthin on cognitive function and neurodegeneration? Do you think the benefits outweigh the risks or vice versa? What are the potential mechanisms or moderators that could explain the variability of the effects across different studies and populations?

4.          Some of the figures are not labeled properly or are too small to read. I recommend revising these figures to make them more clear and readable.

5.          Are there any ethical issues or concerns that you think should be addressed or discussed in the paper? For example, the safety, efficacy, availability, accessibility, affordability, or regulation of astaxanthin.

Reviewer 2 Report

Comments and Suggestions for Authors

The manuscript presents a critical review devoted mainly, but not limited to the effects of astaxanthin on cognitive function and neurodegeneration in humans. The authors claim that there is no critical review outlining the effects of astaxanthin on cognitive function and neurodegeneration holistically in human samples. I would be somewhat skeptical concerning this statement as several reviews of this issue, containing critical elements, have been published relatively recently (e. g. Wu et al. 2015, Grommig et al. 2017, Galasso et al. 2018, Fakhri et al. 2019, El Babhah et al. 2021).

The review is not a systematic review, and the methodology of the choice of data is not presented. Nevertheless, I find this review objective and potentially useful for researchers in the field. Some remarks are given below.

More basic information on astaxanthin, perhaps including the structural formula, would be useful. The authors cite some data indicating that astaxanthin is “ “100-500 times” better than “other antioxidants including vitamin E” (lines 52-53) referring to another review. Such unusually high antioxidant activity of astaxanthin is not reported in most studies so in this respect the review is not especially critical.

Line 49: Why “AX / AST”? Only one acronym can be consequently used in the text

Line 57: pfaffia rhodozyma, please change to Phaffia rhodozyma

Figure 1 seems to be of low quality but it may be due to the low fidelity in pdf formation

Line 143: why to introduce the designation ”(group P)” if it is not used later?

Line 179: “Mean ms:”, rather: “Mean: 359.40 to 363.10 ms”

Line 204: ROS are not classified as “toxic” and non-toxic; their effects depend on the rate of production and removal

Line 222: “a cell line associated with PD”, a misleading statement; this cell line is associated with PD only as a model, after appropriate treatment, not per se

Lines 326-327: “TAC, the measure of Total Antioxidant Capacity”, TAC is not a measure but acronym of Total Antioxidant Capacity.

The text and references should be formatted according to the requirements of the journal.

Comments on the Quality of English Language

I do not have remarks to the language except from words staring from capital letters in the middle of sentences, without any obvious reason
